# Emergent ribozyme behaviors in oxychlorine brines indicate a unique niche for molecular evolution on Mars

Tanner G. Hoog[1], Matthew R. Pawlak [2], Nathaniel J. Gaut [2], Gloria C. Baxter [1], Thomas A. Bethel[3], Katarzyna P. Adamala [1,2] & Aaron E. Engelhart [1,2] ✉

Mars is a particularly attractive candidate among known astronomical objects to potentially host life. Results from space exploration missions have provided insights into Martian geochemistry that indicate oxychlorine species, particularly perchlorate, are ubiquitous features of the Martian geochemical landscape. Perchlorate presents potential obstacles for known forms of life due to its toxicity. However, it can also provide potential benefits, such as producing brines by deliquescence, like those thought to exist on present-day Mars. Here we show perchlorate brines support folding and catalysis of functional RNAs, while inactivating representative protein enzymes. Additionally, we show perchlorate and other oxychlorine species enable ribozyme functions, including homeostasis-like regulatory behavior and ribozyme-catalyzed chlorination of organic molecules. We suggest nucleic acids are uniquely well-suited to hypersaline Martian environments. Furthermore, Martian near- or subsurface oxychlorine brines, and brines found in potential lifeforms, could provide a unique niche for biomolecular evolution.

The possibility of life on Mars has attracted intense interest due to its similarities to Earth, especially during its Noachian period, during which liquid water was thought to be present. At this time, both bodies were rocky planets harboring an ocean world with reducing environments conducive to prebiotic chemistry[1,2]. These similarities have led researchers to speculate that life could have started on Earth and Mars independently. There is a large body of evidence to support this claim, including recent work suggesting wet-dry cycling was present during the late Noachian-Hesperian transition[3,4]. Wet-dry cycles have been invoked as potentially important enabling mechanisms for prebiotic chemical evolution due to their ability to concentrate reaction substrates and drive dehydration-condensation reactions, such as drying of sugars and nucleobases to form nucleosides or clay-promoted RNA polymerization[5–8]. Several missions have identified chemical species on Mars that could support these reactions, such as borates, which

stabilize sugars and were identified at Gale Crater by ChemCam[9], and phyllosilicate clays, which promote RNA polymerization and were identified by the Mars Reconnaissance Orbiter[10].

Despite growing evidence that early Martian conditions were suitable for prebiotic chemistry, obstacles on present-day Mars exist for sustained life into the present Amazonian era. The most notable caveat to life on Mars is the current lack of bulk liquid water. This is due to low temperatures and atmospheric pressure[11], which allowed the Noachian oceans of Mars to dry. A second obstacle to extant Martian life is perchlorate ($ClO_4^-$) and other oxychlorine species which have been shown to be ubiquitous across Mars' surface. Evidence for this from multiple investigations spanning from the Phoenix lander at Green Valley in the Vastitas Borealis artic[12], to the Curiosity rover at Mars' equator (Gale Crater's Aeolis Palus)[13], and even to the analysis of the Martian meteorite EETA 79001 found on Earth[14–16]. Multiple

---

[1]Department of Genetics, Cell Biology, and Development, University of Minnesota, 6-160 Jackson Hall, 321 Church Street SE, Minneapolis, MN 55455, USA. [2]Department of Biochemistry, Molecular Biology and Biophysics, University of Minnesota, 321 Church Street SE, Minneapolis, MN 55455, USA. [3]Department of Ecology, Evolution, and Behavior, University of Minnesota, 140 Gortner Laboratory, 1479 Gortner Avenue, St. Paul, MN 55108, USA. ✉e-mail: enge0213@umn.edu

mechanisms of perchlorate formation have been postulated, with many of these requiring an arid surface to create the perchlorate concentration we see today[17]. However, perchlorate has also been found within samples of the Sheepbed mudstone lacustrine deposit in the Yellowknife Bay of Gale Crater[18]. This sediment dates back to the Noachian-Hesperian era[19], consistent with perchlorate having been a near-constant presence across most of Martian history. This also aligns with proposed mechanisms of perchlorate formation that require aqueous soltions[20,21]. Perchlorate is particularly dangerous due to its toxicity to terrestrial life, which is enhanced by ultraviolet radiation, a third obstacle that bombards Mars' surface[22]. However, these factors can be mitigated. Perchlorate salts exhibit a high degree of hygroscopicity and readily adsorb water from the atmosphere which suggests near-surface perchlorate brines could act as a refuge for extant life[23–25]. Ultraviolet radiation can be substantially reduced by just millimeter thicknesses of regolith, suggesting that the near- or below-surface of Mars could be well protected. Perchlorate brines may provide a stable source of liquid water on present-day Mars, but they also add another problem for habitability: perchlorate solutions are chaotropic and tend to be destabilizing to biomolecules.

Motivated by these reports, and suggestions from investigators that the selection pressures associated with perchlorate brines would drive molecular evolution towards more perchlorate-tolerant organisms[26], we sought to examine how perchlorate brines could have impacted molecular evolution on Mars as a moderately saline environment, such as Noachian oceans[27], transitioned to a hypersaline one, such as recurring slope lineae-associated brines or subglacial reservoirs[25,28]. If Martian life emerged within oxychlorine brines, it would necessarily have addressed this either during prebiotic evolution[29], or as halophilic Martian organisms evolved, as has been observed on Earth[30]. Such molecular adaptations would have been relevant to either scenario. Given that RNA is a particularly salt-tolerant biopolymer, we speculated that it could be well-adapted to perchlorate brines.

In this work, we find that ribozymes have an innate tolerance to perchlorate-like highly evolved thermo- and halophilic proteins, as well as show that oxychlorine brines can support ribozyme activity by enabling homeostasis-like regulatory behaviors and chlorination of organic molecules.

## Results

### Functional RNAs exhibit >10-fold higher tolerance to perchlorate than a mesophilic protein enzyme

Ribozymes, like proteins, fold into intricate three-dimensional structures, imparting the polymer with a specific function. To assess the ability of ribozymes to resist perchlorate denaturation, we examined the impact of perchlorate on a prototypical ribozyme: the self-cleaving hammerhead nuclease (Fig. 1a). We assessed perchlorate-induced changes in the function of this ribozyme in mixed sodium-magnesium chloride and perchlorate solution. We found that the hammerhead ribozyme exhibited a biphasic perchlorate-activity response, with increasing activity up to a maximum at 5 M $NaClO_4$ (apparent first-order rate constant $k_{app} = 0.79$ h$^{-1}$) and retention of function in brines as high as 6-7 M $NaClO_4$ ($k_{app} = 0.24$ h$^{-1}$ for 6 M $ClO_4$; Fig. 1b, c; Supplementary Figs. 1 and 2a). The perchlorate-associated rate optimum occurred at approximately the same concentration of sodium perchlorate as a saturated sodium chloride solution (i.e., 5 M). The self-cleavage rate in chloride solution was approximately twice that of perchlorate solution (1.9-fold higher; 1.48 h$^{-1}$ for 5 M NaCl, Supplementary Fig. 2a). Thus, while high concentrations of perchlorate somewhat suppressed ribozyme function, they still permitted catalysis in solutions nearing the saturation point of sodium perchlorate (9.5 M).

The hammerhead ribozyme consensus sequence we examined was derived from the avocado sunblotch viroid[31,32]. The host organism

that it infects, avocado (*Persea americana*), is mesophilic and well-known to be salt-sensitive, with the well-known Hass cultivar tolerating only sub-millimolar levels of the salinity in irrigation water, even when grafted to salt-tolerant rootstocks[33]. Therefore, we initially compared the hammerhead nuclease to a mesophilic protein nuclease, the restriction enzyme EcoRI (Fig. 1d). In contrast to the hammerhead ribozyme, which exhibits considerable tolerance to perchlorate, EcoRI loses function at perchlorate concentrations ca. one order of magnitude lower (i.e., as low as 0.2–0.5 M, Fig. 1e, f). EcoRI retained only marginal activity in both 0.2 M $NaClO_4$ and NaCl with a 10.7-fold reduced rate of catalysis in perchlorate solutions (Supplementary Fig. 2b; 0.048 h$^{-1}$ for NaCl versus 0.0045 h$^{-1}$ for $NaClO_4$). A second mesophilic protein produced by *E. Coli* was tested. RNase HII, an RNA/DNA hybrid endonuclease, also lost all catalytic activity at just 0.2-0.5 M $NaClO_4$ (Supplementary Fig. 3)[34].

As a second model functional RNA, we examined a representative member of another class: aptamers, which are RNAs that bind specific ligands. The Broccoli aptamer forms a ligand-receptor pair with a small molecule dye, DFHBI-1T, activating its fluorescence upon binding (Fig. 1g)[35]. Broccoli exhibited considerable perchlorate tolerance with retention of function at perchlorate concentrations as high as 5 M $NaClO_4$ (Fig. 1h, i). Remarkably, even at perchlorate concentrations as high as 5 M $NaClO_4$, fluorescent melting analysis of Broccoli/DFHBI-1T reveals sigmoidal traces indicating the proper cooperative folding needed to bind to DFHBI-1T (Supplementary Fig. 4). This retention of function is notable, as the Broccoli aptamer was selected in a mesophilic organismal chassis (*E. coli*). Despite this, the Broccoli aptamer exhibited a more modest response to perchlorate than EcoRI, with 80% retention of activity (as measured by fluorescence) at 2 M perchlorate and 6% at 5 M perchlorate (its upper limit for perchlorate tolerance).

Binding (Broccoli aptamer) and degradative catalysis (hammerhead ribozyme nuclease) represent crucial functions of biology. To better assess the capacity of RNA to assume additional biological roles, we examined a biosynthetic ribozyme: the tC19z ribozyme polymerase[36]. This ribozyme can perform template-directed RNA synthesis, polymerizing single ribonucleotide triphosphates into larger polymers (Fig. 1j). tC19z exhibited gradual diminution of function with increasing perchlorate ($\Delta$yield/d[$ClO_4^-$] = −7.4%/M). The ion dependence of yield in perchlorate was slightly more than half that observed with increasing NaCl ($\Delta$yield/d[$Cl^-$] = −4.1%/M). Indeed, tC19z retained catalytic function at perchlorate concentrations as high as 3–3.5 M (Fig. 1k, l). It only lost its ability to catalyze extension past 7 nucleotides at 3.5 M, with complete loss of activity at 4 M. tC19z, like other functional RNAs tested, was not selected in perchlorate brines, demonstrating that RNA-based functional polymers appear to exhibit innate resistance to the denaturing effects of perchlorate.

### Mesophilic RNAs exhibit capacity to recover from perchlorate-induced denaturation that is similar to extremophilic proteins

EcoRI and RNase HII are two model enzymes from a mesophilic microbial organism (*E. coli*), resembling those more likely to be found in moderate-salt environments that are unlike current Martian conditions. To test model functional biopolymers evolved in conditions more challenging to folding, we examined two protein enzymes from extremophiles found on present-day Earth. As an extremophilic protein analog of EcoRI, we tested the restriction enzyme TaqI-v2 (Fig. 2a), derived from the thermophilic organism *Thermus aquaticus*[37]. We reasoned that the enhanced thermodynamic stability observed in thermophilic biopolymers would impart resilience to perchlorate-induced denaturation. Despite this, TaqI-v2 was remarkably susceptible to perchlorate, with a complete loss of function at 0.2 M $NaClO_4$ (Fig. 2b, c) while still retaining function in 0.2 M NaCl ($k_{app} = 0.42$ h$^{-1}$; Supplementary Fig. 5a).

We next sought to examine a salt-adapted protein enzyme catalyst. We employed the β-lactamase derived from the halophilic

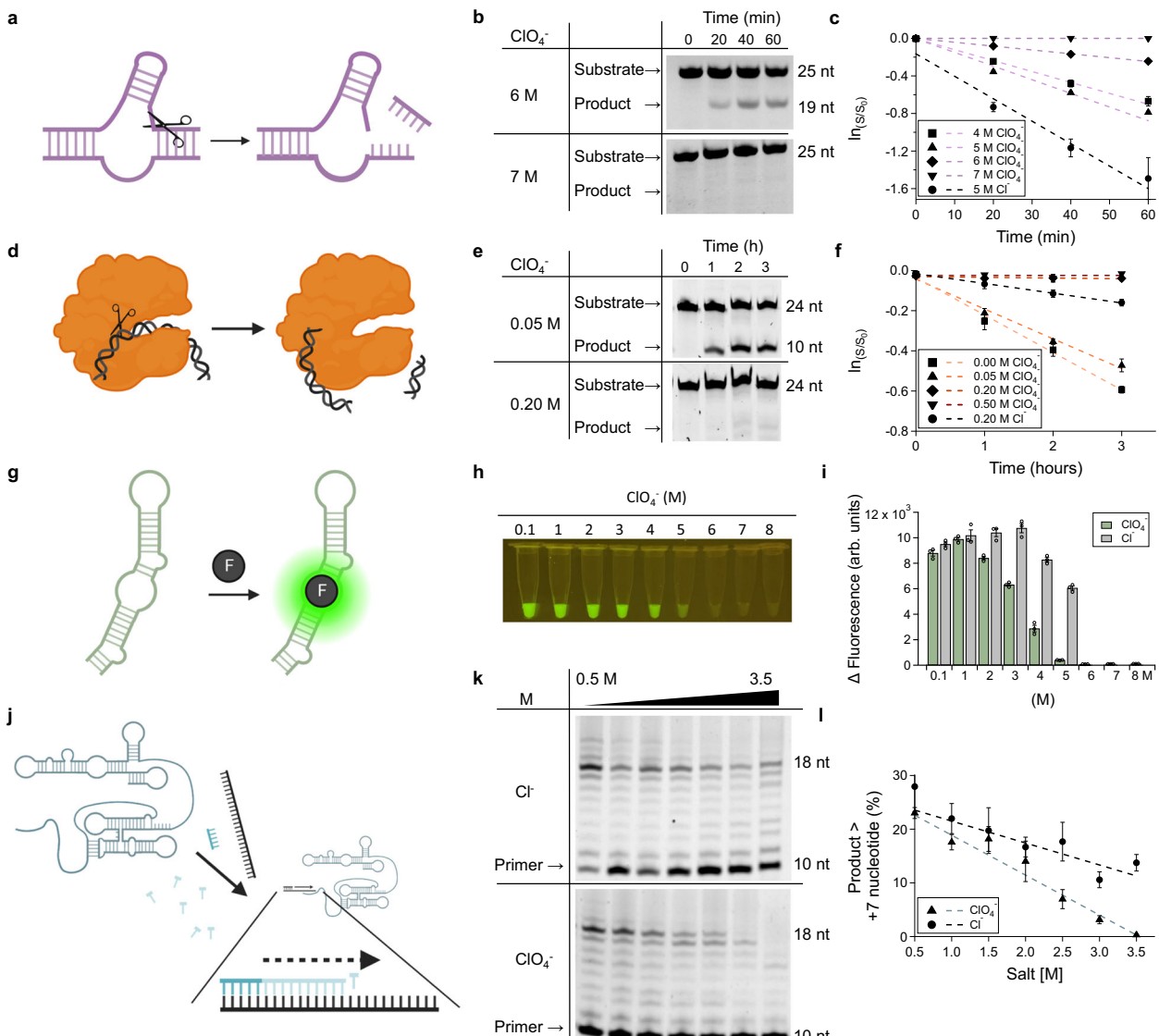

**Fig. 1 | Mesophilic ribozymes are resistant to perchlorate denaturation, while a mesophilic protein is not. a** Cartoon depiction of the hammerhead ribozyme reaction, cleaving one of its component RNA strands. **b** Gel images of the hammerhead self-cleavage kinetic assay (100 nM of each hammerhead strand A and B, 50 mM Tris-HCl buffer pH 8, and 2.5 mM magnesium) in perchlorate brines, showing activity is retained at concentrations as high as 6 M perchlorate. **c** Graphical representation of (**b**) with NaCl control. **d** Cartoon depiction of the EcoRI protein enzyme reaction, cleaving a double-stranded DNA polymer. **e** Gel image of the EcoRI nuclease assay (0.05 U/μL EcoRI, and 1 μM duplex DNA) in perchlorate solutions. **f** Graphical representation of **d** with NaCl control. **g** The Broccoli aptamer binds DFHBI-1T, inducing fluorescence. **h** DFHBI-1T/Broccoli fluorescence in perchlorate brines with 2 μM Broccoli aptamer, 50 μM DFHBI-1T, and 10 mM magnesium. **i** Bar graphs of (**g**) compared to NaCl controls. **j** tC19z ribozyme recruits an RNA template and NTPs to elongate an RNA primer sequence. **k** Gel images of RNA primer extension catalyzed by tc19z in different concentrations of sodium perchlorate/chloride. Reactions consisted of 500 nM tC19z, 500 nM template, 500 nM RNA primer, 4 mM nucleotide triphosphates, 50 mM tris-HCl pH 8.3, 200 mM magnesium. **l** Graphical representation of **k**. Error bars represent the standard error of the mean with $n = 3$ independent experiments. Figure 1a, **g**, and **j** were created with BioRender.com and released under a Creative Commons Attribution-NonCommercial-NoDerivs 4.0 International license.

bacterium *Chromohalobacter* sp. 560 (HaBlap)[38]. Proteins from this genus are known to exhibit adaptations to high salt, including a higher proportion of acidic residues[39]. To assess the impact of perchlorate on the catalytic activity of HaBlap, we monitored β-lactamase activity using nitrocefin, a small molecule that exhibits a color change upon hydrolysis of its β-lactam ring (Fig. 2d). While HaBlap demonstrated the expected resilience to sodium chloride, its activity decreased in the presence of perchlorate, particularly at concentrations ranging from 3 to 4 M NaClO4 (Fig. 2e, f). Despite exhibiting a high degree of halotolerance, the rate of catalysis by HaBlap was still particularly susceptible to perchlorate ($k_{app} = 0.68$ h$^{-1}$ and $k_{app} = 0.12$ h$^{-1}$ at 0 M and 1 M NaClO4 respectively) compared to sodium chloride ($k_{app} = 0.94$ h$^{-1}$ at 1 M NaCl; Supplementary Fig. 5b–d), exhibiting a 7.7-fold decrease in activity at

1 M perchlorate/chloride. Overall, this protein exhibited resistance to perchlorate comparable to levels we observed for ribozymes. This is consistent with a model in which RNAs, with their polyanionic backbones and salt requirements for folding, exhibit innate perchlorate tolerance comparable to that found only in halophilic proteins that have undergone selective pressure and evolution in the presence of high salt.

One model for perchlorate brines on present-day Mars that has attracted intense attention invokes so-called recurring slope lineae (RSL)[25]. It has been speculated that RSL undergoes transient hydration-dehydration cycles. In such cycles, salt concentration would necessarily vary significantly, with high- and low-salt excursions. To examine the tolerance of biopolymers to such cyclic salt concentrations,

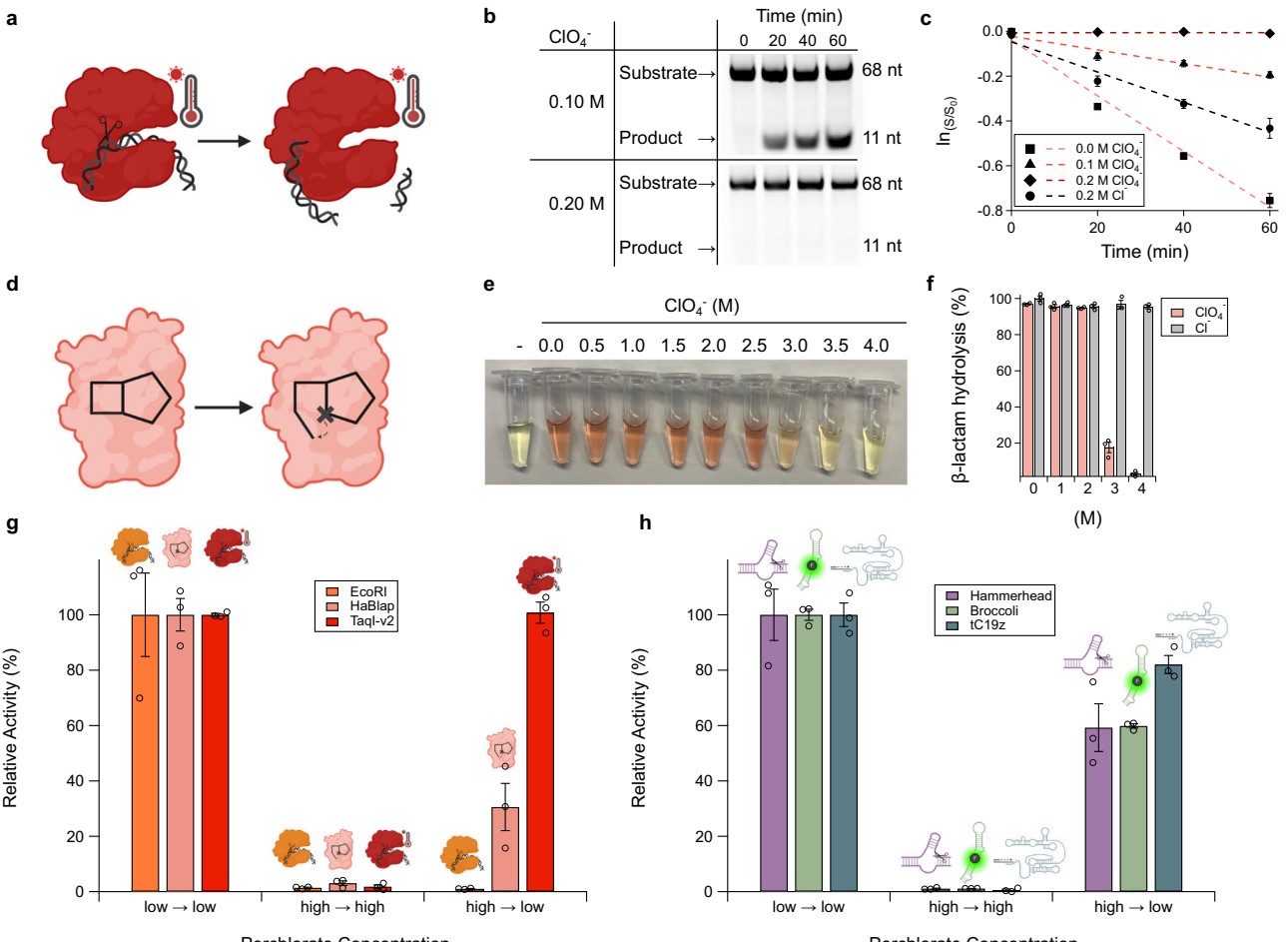

**Fig. 2 | Recovery of activity by functional RNAs from perchlorate-induced denaturation is general, while it is limited to only extremophilic protein enzymes. a** Cartoon depiction of TaqI-v2 nuclease cleaving dsDNA. **b** Gel images of TaqI-v2 nuclease assay (0.05 U/μL EcoRI, and 1 μM duplex DNA) in perchlorate solutions showing the enzyme is inactive at 0.2 M perchlorate. **c** Rate measurements of TaqI-v2 nuclease assay with NaCl control. **d** HaBlap hydrolase opens a β-lactam ring. **e** HaBlap activity assay (5 μM of HaBlap and 50 μM nitrocefin in 5 mM phosphate buffer pH 7) in perchlorate brines performed with the colorimetric β-lactamase substrate nitrocefin. **f** Yields of HaBlap reactions compared to NaCl controls. **g** Activity recovery assay performed on the proteins EcoRI, HaBlap, and TaqI-v2, showing recovery occurs upon dilution from high to low salt only in extremophilic proteins. High and low perchlorate solutions were 5 and 0.05 M, respectively. **h** Activity recovery assay performed on the functional RNAs hammerhead ribozyme, Broccoli aptamer, and tC19z ribozyme, showing recovery occurs upon dilution from high to low salt in all three RNAs. High and low perchlorate solutions were 8 and 0.8 M for hammerhead and Broccoli and 5 and 0.5 M for tC19z. Error bars represent the standard error of the mean with *n* = 3 independent experiments. Figure 2**a**, **d**, **g**, and **h** created with BioRender.com released under a Creative Commons Attribution-NonCommercial-NoDerivs 4.0 International license.

we investigated the reversibility of perchlorate-induced denaturation by examining whether proteins and RNAs could regain functionality after dilution in perchlorate brines. Among those studied, the thermophilic restriction enzyme TaqI-v2 regained all activity upon dilution to low salt, the halophilic β-lactamase HaBlap recovered partial activity (30%), and the mesophilic restriction enzyme EcoRI was irreversibly denatured, with no activity upon dilution to lower salt (Fig. 2g). In contrast, the three RNAs studied uniformly recovered activity upon dilution (Fig. 2h). The Broccoli aptamer and hammerhead ribozyme exhibited ca. 60% restored activity upon dilution and tC19z recovered 82% of its initial activity. Taken together, these results suggest that functional RNAs possess salt tolerance comparable to that of the most extremophilic proteins. This underscores the inherent resilience of RNA to salt, since these RNAs were selected and evolved in conditions comparable to those in which the substantially less salt-tolerant mesophilic proteins evolved.

## Perchlorate enables new regulatory functions for ribozymes
In addition to its role in RSLs on present-day Mars, wet-dry cycling has been proposed as a means of catalyzing biologically relevant prebiotic

reactions, such as polymerization by dehydration-condensation reactions[40]. The wet cycle gathers and solubilizes the necessary components, and the dry cycle concentrates them and removes water produced in the reactions, thereby increasing the rate of reaction. Prompted by recent findings supporting wet-dry cycles on early Mars (late Noachian/early Hesperian)[4], we sought to examine the impact of dilution/concentration cycles of perchlorate brines on ribozyme activity.

A key function required of emergent life is a homeostatic function or regulation of metabolic function in response to environmental conditions. In a model of early Terran life, we previously showed that the hammerhead ribozyme could exhibit a switchable function in the presence of short random hexamers of RNA, as would be found in nonenzymatic RNA polymerization reactions[41]. These hexamers bind to the ribozyme, preventing it from folding and performing its self-cleavage reaction. In this system, we observed that the dilution obtained by the growth of model protocells could activate ribozyme function. Given the lower concentrations of perchlorate thought to exist in the Noachian era on Mars, we speculated that perchlorate could enable a similar role, acting as a nondenaturing counterion at

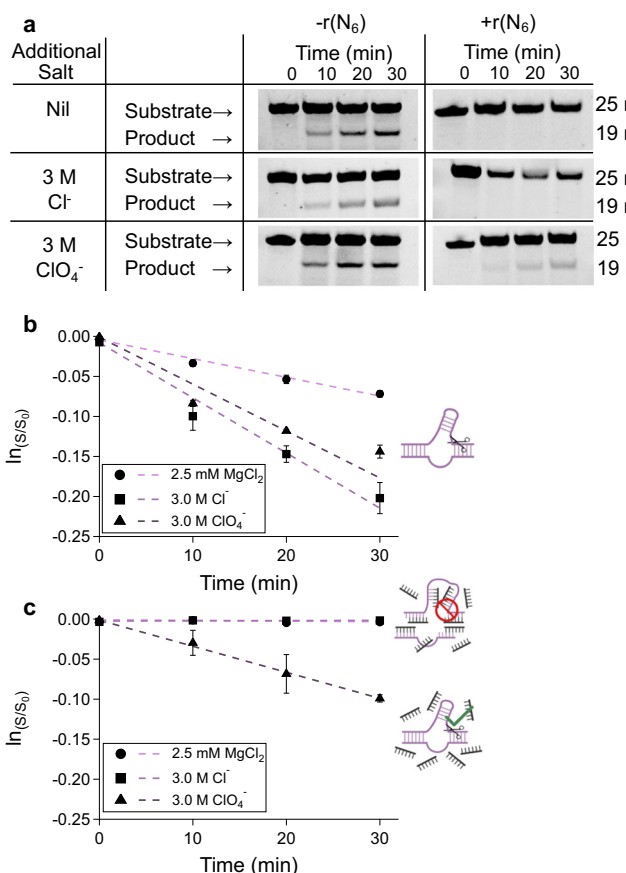

**Fig. 3 | Perchlorate-controlled ribozyme nuclease activity. a** Gel images of hammerhead self-cleavage assay performed in the presence of $rN_6$ and different chaotropic solutions. Reaction conditions consisted of 200 mM Tris-HCl pH 8, 100 nM Hammerhead strand A and B, 250 μM $rN_6$, 2.5 mM magnesium. **b, c** graphical representation of **a** exhibiting perchlorate denaturing catalytically unproductive hammerhead-$rN_6$ complexes, allowing hammerhead renaturation and cleavage. Error bars represent the standard error of the mean with $n = 3$ independent experiments. Figure 3**b, c** was created with BioRender.com released under a Creative Commons Attribution-NonCommercial-NoDerivs 4.0 International license.

low concentrations. At higher but not saturating concentrations, such as those found in a drying pool, they could enable the denaturation of inactive hammerhead-hexamer complexes while still allowing for the formation of active hammerhead. Consistent with our results in the model protocell system, we observed loss of hammerhead function at low salt in the presence of 100 molar equivalents of random hexamers (Fig. 3a–c; $k_{app} = 0.14 \text{ h}^{-1}$ without hexamers, no product detected with hexamers). This trend stayed the same when the Hofmeister-neutral anion chloride was in solution ($k_{app} = 0.39 \text{ h}^{-1}$ without hexamers, no product detected with hexamers). When modest perchlorate was present (3 M), we observed restoration of 69% of ribozyme function by the apparent first-order rate constant ($k_{app} = 0.29 \text{ h}^{-1}$ without hexamers, $0.20 \text{ h}^{-1}$ with hexamers). We also examined hammerhead activity in solutions with the neutral denaturant urea. The hammerhead ribozyme was completely deactivated in the urea solutions with the same molarity used with perchlorate (3 M), and in lower concentrations of urea (1 M) it remained inactive in the presence of interfering hexamers (Supplementary Fig. 6; $k_{app} = 0.04 \text{ h}^{-1}$ without hexamers, no product detected with hexamers). These results show perchlorate's mild denaturing properties for RNA can dissociate interfering oligomers, as might be found in a primitive RNA synthesis reaction, without destroying the ribozyme's native fold. This type of denaturation would be ideal in scenarios such as wet-dry cycles where RNA can exhibit

enhanced function upon concentration in perchlorate brines. This behavior could allow for the dissociation of replication intermediates in copying reactions performed in dilute solution, allowing for switching on of functional ribozymes upon concentration.

## An RNA-heme holoenzyme catalyzes halogenation reactions with the oxychlorine species chlorite

Motivated by the emergent functions enabled by perchlorate, we considered the impact of other oxychlorine species on ribozyme function. By redox potential alone, perchlorate is characterized as a strong oxidizing agent ($NaClO_4/NaClO_3$ $E° = 1.226$ V). However, due to a large electronegative ionic sphere surrounding a kinetically inaccessible chlorine center, this compound is inert to both nucleophilic attack and reduction[42]. Numerous investigators have noted perchlorate's lack of reactivity, with one investigator reporting that a ca. 25-year-old bottle of concentrated 72% $HClO_4$ had maintained its acid titer and showed no sign of chloride buildup over this time[42].

Perchlorate ($ClO_4^-$) is the highest (+7) oxidation state of chlorine, but less oxidized oxychlorine species such as chlorite ($ClO_2^-$, +3) and hypochlorite ($ClO^-$, + 1) are thought to be formed in the Martian chlorine cycle[43]. These oxychlorine species are also much more reactive, exhibiting a higher redox potential than perchlorate ($NaClO_2/NaClO$ $E° = 1.674$ V; $NaClO/Cl_2$ $E° = 1.630$ V respectively). These compounds can perform halogenation reactions with protein enzymes. The horseradish peroxidase (HRP) enzyme, which is best known for its namesake reaction with hydrogen peroxide, can also react with chlorite (Fig. 4a), performing a complete peroxidase cycle with its heme (a porphyrin ring with an iron core) cofactor[44]. In this reaction cycle, chlorine dioxide is formed, which can subsequently chlorinate a range of substrates, such as monochlorodimedon (MCD; Fig. 4b).

Several G quadruplex nucleic acids are well-known to perform HRP-like peroxidase chemistry. A G quadruplex is a secondary structure of RNA composed of stacks of quartets of four guanines coordinating a monovalent cation such as potassium (Fig. 4c). This structure can bind heme (Supplementary Fig. 7a), catalyzing the same peroxidase reactions as HRP, which we have previously demonstrated in high-perchlorate solutions using DNA[45]. We observed that these reactions also occur in the RNA context with the RNA G quadruplex rPS2.M even in near-saturated perchlorate solutions of 8 M (Supplementary Fig. 7b)[46,47]. Motivated by these findings, we hypothesized that this same RNAzyme could also catalyze halogenation reactions.

Accordingly, we assessed the capacity of the rPS2.M/heme holoenzyme to catalyze chlorination reactions with $NaClO_2$ (Fig. 4d). To monitor this reaction, we employed monochlorodimedon, a well-known substrate for enzymatic chlorinations[44]. MCD undergoes a loss of absorption at 290 nm when its 1,3-diketone enolate tautomer is disrupted upon full chlorination of its α-carbon to dichlorodimedon (DCD). As we hypothesized, rPS2.M/heme exhibited the expected loss of MCD absorbance consistent with chlorination (Fig. 4d, e). Analysis of rPS2.M/heme-catalyzed chlorinations by Job's method[48] of continuous variation with varied chlorite/MCD ratios showed the reaction occurred with a stoichiometry of 3 $NaClO_2$:2 MCD (Fig. 4d inset). Notably, this differs from HRP-catalyzed chlorinations, which occurred with a 1:1 stoichiometry (Fig. 4b inset). We hypothesize that this difference in stoichiometry is due to hypochlorous acid accumulating as a byproduct of the peroxidation cycle. Previous research investigating the halogenation reaction with HRP has shown that hypochlorous acid can oxidize heme to Compound I[44], and our results show that HRP is capable of catalyzing peroxidation cycles of heme using only sodium hypochlorite, while rPS2.M/heme does not (Supplementary Fig. 8a). These results indicate that the rPS2.M/heme holoenzyme is mechanistically distinct from HRP in chlorination reactions. Notably, the RNAzyme performed MCD chlorinations faster than HRP, with an initial apparent first-order rate constant of 1.45 $\text{min}^{-1}$, 2.5-fold higher than HRP chlorination ($k_{app} = 0.57 \text{ min}^{-1}$; Fig. 4e). We also found that

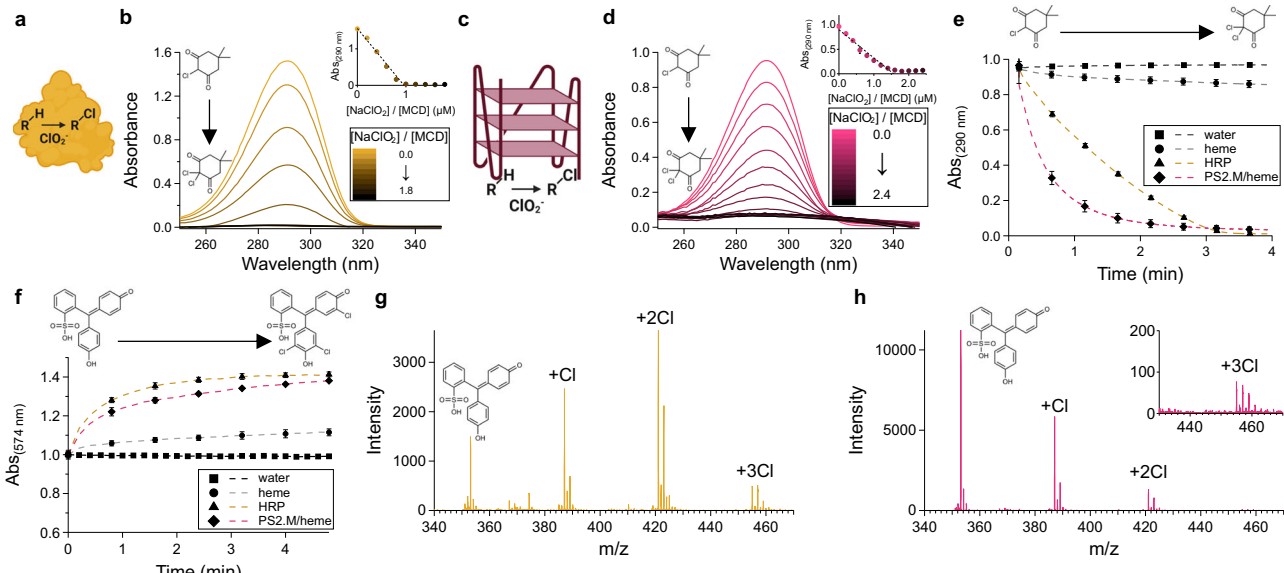

**Fig. 4 | An RNA/heme holoenzyme catalyzes halogenation reactions with chlorite. a** Horseradish peroxidase (HRP) catalyzes a chlorination reaction using chlorite ($ClO_2^-$) as an electron acceptor and chlorine donor. **b** HRP-catalyzed chlorination of monochlorodimedon (MCD) with titrating amounts of $NaClO_2$. **c** A G quadruplex-heme ribozyme (rPS2.M/heme) catalyzes a chlorination reaction using chlorite. **d** rPS2.M/heme-catalyzed chlorination of MCD titrating amounts of chlorite. **e** Kinetic assay of HRP- and rPS2.M/catalyzed MCD chlorination monitored by absorbance at 290 nm. The assay was performed with 5 μM catalyst (HRP, rPS2.M/heme, or just heme), 50 μM MCD, and 100 μM $NaClO_2$ in 100 mM Li-HEPES

pH 7.4. **f** Kinetic assay of phenol red (50 μM) chlorination monitored by change in absorbance at 574 nm catalyzed by HRP or rPS2.M/heme with 200 μM $NaClO_2$. **g, h** Electrospray ionization mass spectrometry of phenol red after chlorination reaction catalyzed by HRP (**g**) and rPS2.M/heme (**h** +3 Cl product shown in zoomed inset). Expected molecular weights: 353.05, 387.01, 420.97, and 454.93. Found molecular weights: 353.1, 387.1, 421.1, and 455.0. Error bars represent the standard error of the mean with $n = 3$ independent experiments. Figure 4**a** was created with BioRender.com and released under a Creative Commons Attribution-NonCommercial-NoDerivs 4.0 International license.

the rPS2.M/heme holoenzyme can withstand up to 5–6 M perchlorate and still catalyze chlorination of MCD compared to heme alone (Supplementary Fig. 8b–d). HRP was also able to withstand perchlorate in the 5–6 M range. This enzyme possesses a network of disulfide bonds stabilizing its fold, providing a potential explanation for its high stability[49].

Electron transfer biocatalysts must all contend with the fact that they perform reactions that are potentially destructive to the catalyst itself[50]. We assessed rPS2.M/heme for the maximum turnover number of the heme cofactor by performing reactions containing excess rPS2.M and $NaClO_2$. In these reactions, we found that heme underwent an average turnover number of 12.5 peroxidation cycles before becoming nonfunctional (Supplementary Fig. 9). Thus, the rPS2.M/heme holoenzyme is a true, multiple-turnover chlorination ribozyme catalyst.

We sought to interrogate possible ribozyme-catalyzed chlorinations further by chlorinating a second substrate, phenol red. Phenol red, a polyaromatic molecule, differs as a chlorination substrate from MCD, a cyclic 1,3-diketone. When monitored by visible spectroscopy, phenol red exhibited a change in its absorption spectrum upon reaction with $NaClO_2$ and either HRP or the PS2.M/heme holoenzyme, consistent with aromatic chlorination and production of chlorophenol red (and chlorination intermediates, Fig. 4f; Supplementary Fig. 10a–d). Analyses of these reactions by electrospray mass spectrometry (ESI-MS) revealed mono-, di-, and trichlorination products from both the rPS2.M/heme holoenzyme and HRP (Fig. 4g, h; Supplementary Fig. 10e, f).

## Discussion

Here, we have shown experimental evidence that demonstrates RNA is uniquely well-suited to function within hypersaline oxychlorine brines like those thought to occur on Mars. This is evident from the fact that (1) functional RNAs, even when selected in low salt, retain function in highly concentrated (3–6 M) solutions of perchlorate; (2) perchlorate

enables emergent ribozyme function due to its property of acting as a mild denaturant for RNA, enabling regulatory behavior; and (3) chlorite allows a G quadruplex/heme holoenzyme complex to perform chlorination of organic substrates. These findings add to the growing number of studies showing that perchlorate is not as deleterious to biomolecular function as was once thought. For example, perchlorate has been shown to increase the activity of α-chymotrypsin at low temperatures by lowering its activation enthalpy[51]. Additionally, pressure can reverse the deleterious effects of perchlorate on biomolecular interactions such as on liquid-liquid phase separation thought to be necessary for compartmentalization of prebiotic life[52], and even the binding of small ligands to tRNA[53].

The observation of oxychlorine species in Martian regolith has led several investigators to seek out perchlorate-tolerant extremophiles in Mars analog environments on Earth, such as the Atacama Desert in Chile and Pilot Valley Basin in the United States[44,54–56]. The most perchlorate-tolerant organism found to date is *Debaryomyces hansenii* which can grow in solutions up to 2.4 M $NaClO_4$[26]. Furthermore, perchlorate-reducing bacteria such as *Dechlorospirillum, Azospira*, and *Dechloromonas* employ perchlorate reductase enzymes using perchlorate as an electron acceptor and transforming it to a more reduced species, such as chloride[57,58]. These organisms evolved in the presence of perchlorate, and their salt tolerance and capacity for employing perchlorate as an electron acceptor is thought to be an evolutionary event that followed the last universal common ancestor (LUCA) on Earth. The survey of the functional landscape of several representative RNAs and proteins in perchlorate brines presented here shows that RNA is innately resistant to perchlorate, that it is more resistant to perchlorate than several mesophilic proteins, and that it possesses this resistance without the requirement for extended periods of selective pressure and evolution to retain function in perchlorate brines.

Ions can impact biopolymer folding by indirect classical Hofmeister effects caused by affecting the water network. They can also have direct interactions with biopolymers, such as anions (like perchlorate)

with hydrophobic residues, which disrupts the hydrophobic collapse that is critical for protein folding. In nature, many halophilic bacteria produce proteins that possess surface negative charge, maximizing their stability in high-salt solution through like-charge repulsion[59]. Our prior work with A-tract containing DNA and bisulfite-catalyzed deamination of cytosine residues is consistent with this model[60]. There we observed enhanced deamination in A-rich sequences with relatively high surface positive charge, consistent with relaxed electrostatic repulsion of the bisulfite anion. Similarly, we have also observed nucleic acid folding within high-salt nonaqueous solvents, such as the deep eutectic solvent formed by a 2:1 molar mixture of urea and choline chloride, or the ionic liquid HMIm-BF$_4$[61]. Given that RNA is a polyanionic molecule, our results agree with a model in which this polymer is innately resilient to perchlorate denaturation (and salt-induced denaturation generally), owing to its polyanionic backbone.

Despite perchlorate's reputation as a strong oxidizer, perchlorate is kinetically inert under the reaction conditions tested, with RNAs ranging from unstructured (polymerization templates), to moderately structured (hammerhead ribozyme, Broccoli aptamer, rPS2.M/heme holoenzyme), to highly structured (tC19z ribozyme polymerase) retaining function. Furthermore, we observed highly specific chemical transformations in our chlorination reactions. The RNAs we have studied show resistance to perchlorate-induced loss of function occurs at concentrations at least 10-fold higher than mesophilic proteins (EcoRI and RNase HII), an innate ability to refold after perchlorate-induced denaturation similar to that of thermophilic proteins (TaqI-v2) and halotolerance equal to or greater than that of halophilic proteins (HaBlap). While the protein and RNA enzymes tested here have been studied under different conditions, with single-turnover conditions that are standard for hammerhead ribozyme and multiple-turnover conditions for the protein enzymes surveyed, the results of this survey are instructive. Given the apparent innate capacity of RNA to resist perchlorate denaturation, we propose that RNA and related molecules, such as DNA or putative pre-RNAs with alternative but still polyanionic backbones, exist within a unique niche of solvent and salt tolerance.

The presence of oxychlorine species during molecular evolution would thus have afforded a powerful evolutionary advantage to RNA-based functional behaviors within multiple milieus throughout Martian history as oxychlorine species started to accumulate[43] to the high concentrations we see today. Such adaptations to these environments could have been relevant to early or late, as well as prebiotic or biotic molecular evolution. In any such scenario, our results, and those found from the studies of extremophilic life on Earth, suggest that Martian molecular evolution in the presence of high concentrations of perchlorate salts would have had a profound impact on the preferred biopolymers, folds, and catalytic strategies employed in Martian (pre) biotic chemistry. Further, our observation that RNA-heme complexes can catalyze carbon-chlorine bond formation extends the chemical repertoire of known ribozymes to include a key synthetic reaction in organic chemistry, suggesting that oxychlorine ion-ribozyme interactions could have played a key role in prebiotic chemistry. A rich suite of chemical transformations is made possible through the agency of cofactors bound to protein enzymes, and our results extend the transformations possible with ribozyme-cofactor complexes. This adds to the growing lines of evidence that as-yet-undiscovered RNA-cofactor pairs could catalyze a wider range of chemistries than currently known[62,63].

Numerous forms of RNA-based metabolism on Earth have been posited as relics of a past RNA world, despite protein having supplanted RNA as the principal functional biopolymer in known biochemical systems. These RNA-based systems include machinery involved in diverse biochemical processes ranging from peptide synthesis (the peptidyl transfer center of the ribosome in which peptide bonds are formed)[64], RNA splicing (Group I and II introns and self-

cleaving ribozymes)[65,66], and transcriptional and translational control of gene expression (riboswitches)[67]. In the case of the ribosome, the evolutionary cost of fully replacing the translational machinery is thought to be so high that a catalyst with only modest rates is still employed across all known biology. However, in the case of introns, RNases, and riboswitches, other protein-based machinery is employed in biological systems with similar functions that appear to be products of later evolution, but it has not fully replaced its putative ancestral analogs. For example, the Group I and II introns coexist alongside the spliceosome, self-cleaving ribozymes, and protein nucleases both occur in extant biological systems, and riboswitches have not been fully supplanted by the numerous protein-based regulatory elements for transcriptional and translational control of gene expression[65,67]. Thus, even in extant biology on Earth, functional biopolymers coexist in competition with one another, and RNA has not fully ceded all its putative ancestral roles to proteins. Regardless of whether perchlorate was present at the advent of a potential Martian biochemistry or a later event in Martian geochemical evolution, we suggest the unique biopolymer folding environment provided by high concentrations of perchlorate would have provided a powerful selection pressure to favor further forms of RNA-based functional behaviors on present-day Mars.

## Methods

### Oligonucleotides
All oligonucleotides were obtained from Integrated DNA Technologies (Coralville, IA) with either standard desalting (DNA) or HPLC purification (RNA). All oligonucleotide sequences can be found in Supplementary Table 1.

### T7 transcription of RNA
Longer RNA molecules were transcribed in-house using T7 RNA polymerase. After polymerization, RNA was isolated by treating with TURBO DNase (Invitrogen, Cat#AM2238) for 30 min at 37 °C, followed by gel purification (10% urea-PAGE).

### Salts and buffers
All salts and buffers used, except where specified, were obtained from Sigma-Aldrich (Milwaukee, WI) and used as received.

### Hammerhead kinetic assay
The hammerhead ribozyme was comprised of two strands (hammerhead strand A/B; see Supplementary Table 1). The reaction conditions were 100 nM hammerhead, 50 mM Tris-HCl buffer pH 8, 2.5 mM MgX$_2$ (X = Cl$^-$ or ClO$_4^-$), and NaCl or NaClO$_4$ added as needed to reach the final chloride or perchlorate concentration specified in the text. To reach near-saturated levels of perchlorate/chloride, hammerhead strand A, and buffer were lyophilized prior to the addition of NaX/ hammerhead strand B. Reactions were started with the addition of magnesium and incubated at 25 °C. Reactions were stopped with the addition of 3 volumes of ethanol and 100x molar equivalents of a DNA complement to strand A to improve electrophoretic separation. The oligonucleotides were isolated by ethanol precipitation. Cleavage was monitored using a 5′-fluorescein label on hammerhead strand A, which was separated on a 20% urea-PAGE gel and visualized using an Omega Lum G Imaging System (Omega Lum Image Capture Software V 2.1.2017.0). Gel images were quantified using GelQuant (GelQuant.NET V 1.8.2; Biochemlabsolutions.com) by analyzing the product as a percentage of all products.

Short hexamer interference assays were performed by first lyophilizing Tris-HCl, hammerhead strand A, and rN$_6$ in the same tube. Reactions were started by adding hammerhead strand B mixed with the appropriate concentration of salt described in the text. The final concentrations were 200 mM Tris-HCl pH 8, 100 nM Hammerhead, 250 μM rN$_6$, 2.5 mM MgX, and variable NaX/urea.

## Restriction enzyme kinetic assay in NaClO₄

EcoRI (New England BioLabs Inc. Cat#R0101S) activity in perchlorate solutions generally consisted of 0.05 U/µL enzyme, 1X EcoRI buffer (New England BioLabs Inc. Cat#B7006S), and 1 µM duplex DNA (see Supplementary Table 1). NaX concentration varied per experiment as noted in the text. Reactions were started by incubating at 37 °C. Reactions were stopped and cleaned up with the addition of 75% ethanol. DNA cleavage was quantified using a 5′-fluorescein label and a 20% urea-PAGE gel.

RNase HII (New England BioLabs Inc. Cat#M0288S) was assessed with the same protocol but in 1X ThermoPol Buffer (New England BioLabs Inc. Cat#B9004S).

TaqI-v2 ((New England BioLabs Inc. Cat#R0149S) was assessed with the same protocol but in 1X rCutSmart Buffer (New England BioLabs Inc. Cat#B6004S) and incubated at 65 °C. DNA was resolved on a 10% urea-PAGE gel.

All gels were analyzed using an Omega Lum G Imaging System using the Omega Lum Image Capture Software V 2.1.2017.0. GelQuant (GelQuant.NET V 1.8.2) was used for quantification.

## Broccoli fluorescence and melting temperature

Samples contained 2 µM Broccoli aptamer, 50 µM DFHBI−1T, 10 mM MgX₂ (X = Cl⁻ or ClO₄⁻), and NaX added to reach the desired levels of perchlorate/chloride (see text) in a buffer solution of 50 mM Li-HEPES pH 7.4 (and 5% DMSO from DFHBI−1T stock). To reach near-saturated levels of perchlorate/chloride, Broccoli oligo, and buffer were lyophilized together prior to resuspension in salt/DFHBI-1T.

The melting traces of the broccoli aptamer were obtained using a BioRad CFX96 Touch Real-Time PCR Detection System using the software BioRad CFX Maestro 2.3 V 5.3.022.1030 measuring fluorescence every 1 °C in two cycles of $15 \to 90 \to 15$ °C. Melting temperatures were calculated by fitting a sigmoidal curve to the melting trace in Igor Pro. Reported fluorescence values in Fig. 1 are from 15 °C of the second heat curve.

## Ribozyme polymerase assay

This assay was performed using a modification of the previously published procedure[36]. Briefly, tC19z and the RNA template strand (see Supplementary Table 1) were annealed by heating to 80 °C for 2 min then cooling to 4 °C for 5 min. Nucleotide triphosphates (NTPs), and Tris-HCl pH 8.3 were mixed in, and the samples were lyophilized. The reaction was started by adding an RNA primer (5-fluorescein labeled) and salt solution to the lyophilized samples and incubated at 4 °C for 1 week. The final concentrations of the reaction were 500 nM tC19z, 500 nM template, 500 nM RNA primer, 4 mM NTPs, 50 mM Tris-HCl pH 8.3, 200 mM MgX₂ (X = Cl⁻ or ClO₄⁻), and NaX was added to reach the perchlorate/chloride concentration described in the text. After incubation, the RNA was ethanol precipitated. The extended primer products were displaced from the RNA template by adding 100x molar equivalents of the template complement strand (DNA) and heated to 80 °C for 4 min. Extended primer products were resolved on a 20% urea-PAGE and visualized using a 5′-fluorescein labeled primer. Gels were imaged using an Omega Lum G Imaging System using Omega Lum Image Capture Software V 2.1.2017.0. Quantification of each band was obtained using GelQuant software (GelQuant.NET V 1.8.2).

## β-lactamase (HaBlap) activity assay

The β-lactamase plasmid (pExp-Bla, Addgene plasmid #112561) was expressed in *E. coli* and isolated using nickel agarose beads (Goldbio, Cat# H-350-5). 5 µM of HaBlap was mixed with 50 µM nitrocefin in 5 mM phosphate buffer pH 7. Perchlorate salt concentration varied by experiment. Nitrocefin hydrolysis was monitored on a SpectraMax 340PC384 plate reader (using SoftMax Pro software V 5.4; Molecular Devices) measuring absorbance at 486 nm and blanked to identical

solutions lacking nitrocefin. Kinetic assays were performed the same way but with 5 nM HaBlap.

High-salt concentrations affected the extinction coefficient of hydrolyzed nitrocefin, which distorted the absorbance readings. We corrected the absorbance readings by incubating nitrocefin with 5 nM HaBlap in buffered solution for 30 min (to fully hydrolyze the nitrocefin) and then diluting that reaction into an array of sodium chloride/perchlorate solutions (nitrocefin final concentration: 50 µM). The extinction coefficient was calculated for each salt solution based on the Beer-Lambert law and graphed (Supplementary Fig. 11).

## Perchlorate recovery assays

All perchlorate recovery assays were performed by incubating the functional RNA/enzyme in an initial buffered molar perchlorate solution for 30 min at 4 °C before diluting into a working solution containing the necessary components for their individual assays (see individual assay requirements above). All recovery assays were normalized to a positive control (low to low samples, Fig. 2g, h) and compared to a negative control (high to high samples, Fig. 2g, h).

**Functional RNAs.** Since the hammerhead ribozyme and Broccoli aptamer were the most tolerant of perchlorate, they received a starting concentration of 8 M perchlorate and an ending concentration of 0.8 M perchlorate (10-fold dilution). The tC19Z ribozyme was less salt tolerant and started in 5 M perchlorate and was diluted 10-fold to 0.5 M perchlorate. The hammerhead ribozyme was initially incubated with only NaClO₄ and 2.5 mM Mg(ClO₄)₂ was added to start the reaction. The Broccoli aptamer dilution/working solution contained DFHBI-1T. The tC19z ribozyme was initially incubated with the template and primer strand and the dilution solution provided the NTPs to the reaction. All positive control initial solutions contained 0.8 M perchlorate and diluted into a working solution also containing 0.8 M perchlorate. The negative controls were the opposite: the initial solution starting concentration was 8 M perchlorate and the working solution also contained 8 M perchlorate.

**Proteins.** Since all tested proteins were relatively susceptible to perchlorate-induced inactivation, 5 M perchlorate was used as the initial concentration and diluted 100-fold to 0.05 M. For the EcoRI and TaqI-v2 assays the dilution solution contained their respective DNA substrate (see Supplementary Table 1). The HaBlap working solution contained nitrocefin. Positive and negative controls were set up like those for the functional RNAs but with 0.05 M and 5 M perchlorate respectively.

## Chlorination assays

**Kinetic assays.** Working solutions generally consisted of 50 µM monochlorodimedone (MCD), 10 µM G quadruplex RNA (rPS2.M), 5 µM hemin, 2.5% DMSO, 100 mM KCl, and 100 µM NaClO₂ in a buffered solution of 100 mM Li-HEPES pH 7.4. To keep the concentration of heme consistent, 5 µM horseradish peroxidase (HRP) was used as a control. The G quadruplex first underwent a heat-cool cycle to attain the correct configuration: heat to 80 °C and cool to 10 °C at 1-min intervals every 10 °C. Then the folded rPS2.M was incubated with hemin for 10 min on ice. The assay was started with the addition of NaX (X = ClO⁻ or ClO₂⁻) and monitored using a Cary 60 UV-Visible spectrophotometer (Agilent Technologies; Cary WinUV ADL Shell Application V5.0.0.1008). Data was referenced to a blank solution containing all reaction components except MCD and NaX.

Phenol red chlorination kinetic assays were performed in the same conditions as those used for MCD but with 50 µM phenol red and 200 µM NaClO₂.

## Electrospray ionization mass spectrometry

Phenol red was isolated from the chlorination assays by running the reaction through a Sep-Pak Plus C18 column (Waters Corporation, Cat#WAT020515), washing with water, and eluting with 35% methanol. Samples were desiccated and resuspended in 100% methanol. Samples were analyzed on a Bruker BioTOF II/TOF-MS in negative mode and 2000 spectra were collected using the TOF Control V 2.2 software.

## G quadruplex/hemin redox assay

The rPS2.M G quadruplex was heat-cycled to with $NaClO_4$ to attain proper configuration: heat to 80 °C and cool to 10 °C at 1-min intervals every 10 °C. The sample was then incubated with hemin at room temperature for 5 min. The G quadruplex/hemin RNAzyme was allowed to react with 10-Acetyl-3,7-dihydroxyphenoxazine (Amplex Red, Cayman Chemical Company, Cat#10010469) and Hydrogen peroxide for 5 min. Fluorescence readings were taken on a SpectraMax Gemini EM microplate reader (Molecular Devices; SoftMax Pro software V 5.4) with an excitation wavelength of 565 nm and an emission wavelength of 585 nm. Data was normalized to a negative control containing Amplex Red but no rPS2.M/hemin. Working solutions contained 5 μM rPS2.M, 1 μM hemin, 5 mM phosphate buffer pH 7.4, 5 μM Amplex Red, 10 μM $H_2O_2$, 0.5% DMSO, and variable $NaClO_4$.

## Data analysis

Data analysis was performed in Igor Pro 9.00.

## Reporting summary

Further information on research design is available in the Nature Portfolio Reporting Summary linked to this article.

# Data availability

The raw data generated in this study are provided in the Source Data file. Source data are provided in this paper.

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

## Acknowledgements

We thank Loren Williams and members of the Adamala and Engelhart laboratories for helpful discussions. This work was supported by NASA Contract 80NSSC18K1139 under the Center for Origin of Life (to A.E.E. and K.P.A.), Research Corporation for Science Advancement-Scialog Award 28754 (to A.E.E.), Heising-Simons Foundation-Scialog Award 2021-3123 (to A.E.E.), and National Science Foundation Award 2123465 (to K.P.A.). Diagrams in figures were created with BioRender.com.

## Author contributions

T.G.H., M.R.P., N.J.G., G.C.B., T.A.B. performed experiments. T.G.H. analyzed and prepared all figures. K.P.A. provided ideas and discussion. T.G.H. and A.E.E. designed the project and wrote the manuscript.

## Competing interests

The authors declare no competing interests.
