## [Peer Review File · Nature Communications]

Emergent ribozyme behaviors in oxychlorine brines indicate a unique niche for molecular evolution on MarsREVIEWER COMMENTS

Reviewer #1 (Remarks to the Author):

The authors explore a fascinating “What if?” scenario related to the origins and early evolution of life on Mars and potentially other worlds. They make a strong case for hypersaline and subsaturating oxychlorine brines based on spacecraft observations of the modern Martian surface. Of note, there is a very strong push in the Astrobiology field to use realistic geochemical conditions in prebiotic chemistry studies; hence, their grounding in realistic Mars surface conditions is a major strength.

The study demonstrates retention of several RNA functions at perchlorate concentrations that are denaturing for several protein enzymes, and they speculate that the anionic nature of RNA may contribute to this retention of activity. The comparison with a protein from a hypersaline-adapted organism is nice. The authors also demonstrate chlorination of two organic species by heme-binding ‘peroxidase’ RNAs, analogous to (but not identical to) reactions catalyzed by horseradish peroxidase (HRP).

I really like this study. Although there are many comments below regarding specifics, most are intended to improve communication and resolve ambiguities. Overall, the study is fascinating, timely, and well-written (with some minor exceptions, below). The data support the conclusions and will stimulate further investigations. I believe that this study will be well regarded and that it can readily be modified for publication in Nature Communications.

MAJOR POINTS

1. The presence of oxychlorine and other oxidized species on modern Mars represents, in part, an increase in the oxidation state of that planet over time, as water molecules were photolyzed and hydrogen was lost to space, leaving behind atomic and molecular oxygen. Authors should make a stronger case for how and when oxychlorine species are thought to have accumulated, especially relative to the Noachian-Hesperian transition. If they were minor or inconsequential species in early stages of the evolution of the Martian surface, then the study loses some of its relevance.

2. The observation of chlorate-dependent ribozyme polymerase readthrough of a short stem-loop structure and relief of repression by random oligonucleotides could both be explained by weak denaturation. As such, similar effects as those observed here might also be observed for 1-3M urea or for tetraalkylammonium salts (and other chaotropic salts).

ALSO IMPORTANT

3. Last paragraph of Introduction. The sentence beginning “Martian life would necessarily have addressed this” needs to be rephrased so as not to presuppose that such life ever existed. One solution (among many): “If Martian life emerged within oxychlorine brines, it would necessarily have had to address these physicochemical realities.” [Separately, the end of that sentence introduces ambiguities. Suggestion: “...or as halophilic Martian organisms evolved, like those that emerged on Earth.”]

4. In numerous places, authors use phrases such as “retain robust activity” or “shows significant activity,” but the criteria separating those qualitative statements from “shows only weak activity” is not given. Where is the threshold separating the various ‘bins’? How well separated are they from the negative control(s)?

5. Authors do a very nice job demonstrating that two proteins are denatured at relatively low concentrations of perchlorate. However, they should be careful not to imply that $n = 2$ allows extrapolation to all mesophilic proteins (e.g., Results, 2nd paragraph, top of p3 “To further validate...” and top of p5).

6. p3 – Sigmoidal traces do not indicate “proper folding,” as stated, but rather ‘cooperative folding.’ It is safe to say that these are still folded molecules.

7. Fig 1k. Main polymerization product appears to be at +8 relative to primer. y-axis of next panel (Fig 1l) refers to “Product >7nt”. Primer itself appears to be 10nt, with 18 additional nucleotides available for polymerization (from Extended Table 1). Are there additional polymerization products beyond +8 (gel image is cut off above that point)? At a minimum, address these points in the text; probably also prudent to show uncropped or less cropped

image in SI.

8. Fig 3 shows readthrough of a short (3bp) hairpin by polymerase ribozyme in the presence of chlorate. Two points: a) Have the authors looked at longer hairpins or at stem-loops with different stabilities (e.g., one or more GC pairs or capped by a stable tetraloop)? What is the upper limit for this helicase-like effect? b) Graphics above panels e and f are misleading. They look like two different templates are used in the two experiments, when the intention is to depict unwinding of the inhibitory 3bp 'roadblock.' Please adjust image or state explicitly in the legend what the graphics represent.

9. p9, bottom. To provide context for understanding the HRP-catalyzed reaction, state (and show data for) chlorite concentration that denatures HRP. Optional: Evaluation of HRP stability in chlorite could be shown in conjunction with protein denaturation in Fig 1, either as additional panels for Fig 1 or within SI.

10. Fig 4e, these reactions were performed in the presence of chlorate, correct? Give concentration.

11. Turnover calculation is based on data shown in ED Fig 6. Indicate how "50 uM MCD" is calculated from these data; if different data were used to arrive at that value, then show the data from which that number is calculated.

12. ED Fig 7 does not communicate well. Panels (a) and (b) appear identical, except for (b) being slightly fuzzier, making it difficult to discern a meaningful interpretation. In panel d, point to expected product (near 381-383 units) as in Fig 4i-j and give expected mass.

MINOR DETAILS

- When the object of a verb is a clause, the clause needs to be introduced by "that." For example, "I know THAT this is true." rather than "I know this is true." ("This is true" is a clause.) Some examples from this manuscript:
- Second line of abstract: "indicating that oxychlorine species"

- Sixth line of Intro: “suggesting that near-surface”
 - Fifth line of Results: “We found that the hammerhead”
 - Third paragraph of “An RNA-heme holoenzyme” section, fifth line: “We observed that these reactions”
 - Near end of next paragraph: “These results indicate that the rPS2.M/heme holoenzyme”
 - Near end of second paragraph of Discussion: “our results show that RNA is innately resistant”
 - Two paragraphs later: “The RNAs we have studied show that resistance to”
- (Note that penultimate paragraph already uses this more proper structure: “In any such scenario, our results ... suggest that Martian molecular evolution...”)
- Second sentence of last paragraph of Introduction needs a verb. (Starts “Particularly, ...”)

p2, Results, first paragraph. Hyphenate “first-order”. Suggestion: Remove comma and reword as “and retention of function in brines...” Suggestion for last sentence: “Thus, while the highest concentrations of perchlorate somewhat suppressed ribozyme function, robust catalysis was still permitted at remarkably high concentrations.”

p2, Results, second paragraph. First two sentences: “...we examined was derived from the avocado sunblotch viroid. The host organism that it infects, avocado (*Persea americana*) is mesophilic and is among the most salt-sensitive crops, ...”

p3, second full paragraph. Suggestion: “Ligand binding (exemplified by Broccoli aptamer) and ...” Five lines later: Odd phrasing in “This was slightly more than half the yield,” since the preceding sentence was about $d(\text{yield})/d[\text{anion}]$. “This” is ambiguous.

Fig 1 legend. Capitalize “Graphical” in text for panel f.

p5, near the bottom. Suggestion: “In contrast, all three RNAs studied recovered activity upon dilution...” (It took me a while to realize that the three bars represent three different RNAs rather than a progressive dilution from high chlorate to low chlorate. Explicitly saying “three” could help with this.)

p7, second paragraph. Use plural (“hexamers”) throughout, since there are 4096 of them.

p9, second paragraph. Show charges for chlorite and hypochlorite; these are currently shown as neutral species! Cite reference for “Job’s method.”

p10, top. Use singular: “To our knowledge, this is the first report of carbon-chlorine formation...”

p11, Discussion. Suggestion for first sentence: “...evidence demonstrating that RNA is uniquely well-suited ... brines (no comma) like those thought to occur...” 2nd and 3rd paragraphs: Add comma before WHICH: (“which can grow in solutions...” and “which disrupts the hydrophobic collapse...” End of 2nd paragraph: It would safe to omit “periods.” 4th paragraph, change “like thermophilic proteins” to “similar to that of ...”

p18 Methods. Hammerhead Kinetic Assay. Is 75% ethanol the final concentration or (as written) the concentration of the material that was added? Next paragraph: Were strands lyophilized together or separately?

p19. Ribozyme polymerase assay. Optional: It would be safe to omit “out of solution.” Two paragraphs later, add comma after “nitrocefin” in first line. Next paragraph, specify what samples were used as “positive control and negative control”

EXTENDED DATA: Define meanings of error bars and number of independent observations (n) on which they are based.

ED Fig 3. What equation was used to fit these data? Indicate where the inferred midpoint transitions (and T_m values) lie. It is difficult to infer an accurate T_m from only 5 data points! Even if these data were not used to calculate T_m values (which appears to be the case), it looks like the data points are simply connected with a cubic spline so that the curves pass exactly through the points.

ED Fig 5. In x-axis, denominator needs parentheses

Reviewer #2 (Remarks to the Author):

Hoog et al. present a survey of the comparative performance of functional RNAs and proteins in perchlorate brines, as a model for functional biomolecules in similar brines that could form on the surface or sub-surface of Mars in the latter dry stages of that planet's geological evolution. Functional RNAs investigated include the hammerhead ribozyme, Spinach fluorogenic aptamer, and tC19Z RNA polymerase ribozyme. The proteins investigated include nucleases EcoRI, RNaseHIII, and TaqI-v2, the former two from mesophilic *E. coli* bacteria, the latter from the thermophilic *T. aq.* bacteria, and a β -lactamase from a halophilic bacterium.

The authors reach the conclusion that RNA is broadly and generally tolerant of concentrated perchlorate, with each RNA tested able to tolerate at least 2 M and as much as 6 M perchlorate. Proteins, in contrast, cannot tolerate perchlorate above 0.5 M, with the exception of a halophile derived β -lactamase. The authors ascribe this difference, primarily, to the polyanionic nature of RNA that would be resistant to salt-induced denaturation, while proteins must evolve negatively charged surfaces or extreme stability, as occurs in halophilic or thermophilic proteins, respectively, to achieve comparable levels of resistance.

The ability of RNA to maintain function in concentrated perchlorate is shown to enable further advantages. Complex mixtures of RNA oligonucleotides are disruptive to RNA function, by binding of oligonucleotides to functional regions of an RNA structure. In a model of such disruption using the hammerhead ribozyme, it is shown that concentrated perchlorate can significantly reduce this inactivation, presumably by destabilizing interstrand RNA pairing. Likewise, modest levels of perchlorate appear to improve polymerase read-through of an RNA template with a central stem-loop structure that would normally block the polymerase, suggesting that mild denaturation of RNA structure in perchlorate brines could improve RNA-catalyzed replication of structured RNA.

Finally, and separately, the horseradish peroxidase protein enzyme and the rPS2.M ribozyme, both of which use heme as a catalytic cofactor, are shown to chlorinate various compounds from chlorite, another known component of Martian soils, at comparable rates. Chlorination appears to be regiospecific, can occur as a multi-turnover reaction, and can produce polychlorinated products. These results are taken as preliminary evidence that

either biopolymer could evolve to incorporate environmental oxychlorine into metabolism.

Overall, the authors present compelling, albeit preliminary, evidence that RNA is intrinsically better suited to maintain function in concentrated perchlorate solutions, and, indeed, that these brines may offer specific advantages for RNA catalysis. The authors hypothesize that functional RNA would have been favored by evolution in perchlorate brines over functional proteins, either as an emergent form of life from prebiotic chemistry or an adaptive response of extant Martian life as the Martian surface dried to its current state. This is a highly valuable approach to origin of life investigations, tying biochemical behavior to prebiotic environments on Earth and elsewhere, in this case one that is decidedly hostile to extant biology, and thus not explored in detail by biochemists focused on the biological behavior of macromolecules. These results will be of general interest to RNA biochemists, prebiotic chemists, and astrobiological fields more broadly, and is highly suitable for publication in Nature Communications.

There are three points that I think should be addressed before publication, however. Each can be addressed by minor revisions to the text alone, although additional experimental work may increase the confidence of some conclusions. First, nuclease activity is the only shared enzyme type for both protein and RNA enzymes in this survey. However, it is unclear if these were assayed under comparable kinetic regimes. The hammerhead ribozyme was assayed under more-or-less single turnover kinetics, with the two pieces of the ribozyme (one strand of which is the substrate RNA that is cleaved) present in equimolar amounts. It is unclear if the protein nucleases were assayed under similar conditions, as the enzyme input to these reactions is given as units of activity, not molarity. It is sometimes difficult to determine protein concentration in enzyme stocks supplied commercially, but it should be possible for at least some of these enzymes. The manuscript should clarify if the kinetic regimes for RNA and protein were similar, and if not, the implications for differing effects due to different rate-limiting steps: bond cleavage vs product on/off rates etc. As a broad survey of activity, a perfect match between kinetics isn't required, but the consequences of any differences should be discussed. It would also be helpful if some basic details of the kinetics (enzyme and substrate stoichiometry, pH or temperature where relevant) for each reaction were provided in the results or relevant figure legends, rather than only in the

methods.

Second, a relatively minor point, but the difference in recovery of activity after exposure to high perchlorate concentrations is compared between the hammerhead ribozyme, broccoli aptamer, and tC19Z polymerase ribozyme in the last paragraph of page 5. The greater recovery for the polymerase may be illusory. From the methods, it appears that the polymerase ribozyme was assayed by yield at a single time point, rather than measuring an observed rate constant, as was performed for the hammerhead ribozyme. If the reaction is near its maximum extent when the timepoint is taken, rather than in the linear phase, differences in yield from different conditions can be lower than difference in the underlying rate of reaction, as faster reactions will be approaching the maximum extent limit. All three RNAs recover significantly after exposure to perchlorate brines, but the text should not imply that the greater measured recovery for the polymerase is notable, unless a more detailed measurement of polymerase rates is performed.

Third, the fact that perchlorate brines can “loosen” RNA structure, while enabling RNA enzymes to retain function, is extremely interesting, and as noted, important for the copying of structured RNA by polymerase ribozymes. The results reported compare yield of extension products past a 3 nucleotide stem loop in an RNA template in chloride or perchlorate solutions. Sodium is known to be detrimental to the polymerase (J. Attwater, et al., Chemical fidelity of an RNA polymerase ribozyme. *Chem. Sci.* 4, 2804–2814 (2013)), and polymerases can perform idiosyncratically with different templates under different conditions. Also, the sequence of the template stem is AU rich, which is generally copied less efficiently by the polymerase. Although it seems likely that the better performance in perchlorate is due specifically to disruption of the stem loop, it could alternatively be due to perchlorate ameliorating the effects of sodium, aiding in AU copying, or other challenges. Experiments with other templates, either varying stem loop sequence, or including a negative control where the stem loop is disrupted but the ~7 nt portion copied by the polymerase is the same sequence, would provide a much more convincing demonstration that perchlorate is specifically aiding RNA copying by disrupting secondary structure in the template. In the absence of that data, since this is not a central experiment to the paper, it would be sufficient to change the language to make clear the tentative nature of the

conclusions regarding the effects of perchlorate on secondary structure.

Reviewer #3 (Remarks to the Author):

Emergent functional behaviors of ribozymes in oxychlorine brines indicate Mars could host a unique niche for molecular evolution

Hoog TG et al.

Major comments

The paper presents some very interesting work on the effects of perchlorate on biomolecular function. In particular the description of new capabilities associated with nucleic acids is of great interest. The paper is well-written and very clear. I only have some minor comments.

One point: It's a bit strange that the paper doesn't cite any of the previous papers that present information on the effects of perchlorate brines on biomolecular function relevant to Mars. There are these papers for example:

<https://pubs.acs.org/doi/abs/10.1021/jacs.1c01832>

<https://www.nature.com/articles/s41598-021-95997-2>

<https://www.nature.com/articles/s42003-020-01279-4>

Although the work is different, the field of research that deals with perchlorate brines on Mars and effects on biochemical function is very niche and it would be worth acknowledging these, especially those that show improvements in function in perchlorate (for example pressure reversal of deleterious perchlorate effects).

Minor comments

Page 5, line 3. I don't think it is necessary to point out that *E. coli* is terran – could be removed.

Page 5, line 3. It's a bit of a stretch to say that these enzymes could have been found in a Noachian/Hesperian organism as we don't know if Mars ever had life. Maybe this could be toned down to simply say that these enzymes could be plausible universal analogues of any organism.

Page 7, line 4 up. The statement that increases in salt concentrations did not result in

further elongation product, but there was still an increase in elongated could be clearer.

Page 9, line 7 up. Delete period and extra space after 'chlorination reactions'.

Under 'restriction enzyme kinetic assay. Line 9 down. Remove extraneous parenthesis.

Figure 4 is quite dense. Maybe some of the more methodological parts could go in supplemental?

Author responses are shown in green.

REVIEWER COMMENTS

Author response: We gratefully acknowledge the reviewers for the time and effort involved in their careful evaluation of our manuscript and preparation of their suggestions. We have revised the manuscript with changes to the text and additional experiments, which we detail in our response to the reviewers.

Reviewer #1 (Remarks to the Author):

The authors explore a fascinating “What if?” scenario related to the origins and early evolution of life on Mars and potentially other worlds. They make a strong case for hypersaline and subsaturating oxychlorine brines based on spacecraft observations of the modern Martian surface. Of note, there is a very strong push in the Astrobiology field to use realistic geochemical conditions in prebiotic chemistry studies; hence, their grounding in realistic Mars surface conditions is a major strength.

The study demonstrates retention of several RNA functions at perchlorate concentrations that are denaturing for several protein enzymes, and they speculate that the anionic nature of RNA may contribute to this retention of activity. The comparison with a protein from a hypersaline-adapted organism is nice. The authors also demonstrate chlorination of two organic species by heme-binding ‘peroxidase’ RNAs, analogous to (but not identical to) reactions catalyzed by horseradish peroxidase (HRP).

I really like this study. Although there are many comments below regarding specifics, most are intended to improve communication and resolve ambiguities. Overall, the study is fascinating, timely, and well-written (with some minor exceptions, below). The data support the conclusions and will stimulate further investigations. I believe that this study will be well regarded and that it can readily be modified for publication in Nature Communications.

Author response: We are pleased that the reviewer shares our enthusiasm about the biochemical systems we have investigated and the precise geochemical scenario we propose to motivate it. We additionally are pleased they find that our data support our conclusions and believe it will prompt additional follow-on work. This is our hope as well. We have carefully revised our manuscript in response to the reviewers’ suggestions, and we have provided our responses in-line.

MAJOR POINTS

1. The presence of oxychlorine and other oxidized species on modern Mars represents, in part, an increase in the oxidation state of that planet over time, as water molecules were photolyzed and hydrogen was lost to space, leaving behind atomic and molecular oxygen. Authors should make a stronger case for how and when oxychlorine species are thought to have accumulated, especially relative to the Noachian-Hesperian

transition. If they were minor or inconsequential species in early stages of the evolution of the Martian surface, then the study loses some of its relevance.

Author response: We thank the reviewer for the suggestion to strengthen the motivation for the use of oxychlorine species in our work. We have done so in our revision. There are many ways oxychlorine species can form on Mars based on both laboratory investigations (e.g., in model studies of titanium oxide and chloride in the presence of UV light) and studies of analog environments (e.g., the Atacama desert). Additionally, we note that perchlorate has been observed within the Sheepbed mudstone deposit in Yellowknife bay of Gale Crater, which dates to the Noachian-Hesperian transition. In the introduction to our revision, we have cited work demonstrating these phenomena, which are consistent with perchlorate having been a constant presence throughout Martian history.

2. The observation of chlorate-dependent ribozyme polymerase readthrough of a short stem-loop structure and relief of repression by random oligonucleotides could both be explained by weak denaturation. As such, similar effects as those observed here might also be observed for 1-3M urea or for tetraalkylammonium salts (and other chaotropic salts).

Author response: We thank the reviewer for this suggestion. In response, we have performed hammerhead/rN6 assays in the presence of urea. These data are shown in Supplementary Figure 6. We started with 3 M urea and found that these conditions prevented hammerhead from performing its catalytic function at all. At a lower concentration (1 M) of urea, we found that the hammerhead ribozyme was active without rN6. However, in the presence of rN6, this was an insufficient denaturant to relieve rN6-associated inhibition of the ribozyme. In the revision, as discussed in point 8 of our response, we have refocused this section on rN6/hammerhead inhibition.

ALSO IMPORTANT

3. Last paragraph of Introduction. The sentence beginning “Martian life would necessarily have addressed this” needs to be rephrased so as not to presuppose that such life ever existed. One solution (among many): “If Martian life emerged within oxychlorine brines, it would necessarily have had to address these physicochemical realities.” [Separately, the end of that sentence introduces ambiguities. Suggestion: “...or as halophilic Martian organisms evolved, like those that emerged on Earth.”]

Author response: We thank the reviewer for the suggestion and in our revision, we have revised this section to not express as high a degree of certitude regarding Martian life as was present in our original submission.

4. In numerous places, authors use phrases such as “retain robust activity” or “shows significant activity,” but the criteria separating those qualitative statements from “shows only weak activity” is not given. Where is the threshold separating the various ‘bins’? How well separated are they from the negative control(s)?

Author response: We thank the author for this inquiry and in response, we have rephrased the language in this section. In our revision, we describe the activity with respect to perchlorate concentration with reference to a saturated solution (9.5 M) of sodium perchlorate to better clarify the concentrations being compared.

5. Authors do a very nice job demonstrating that two proteins are denatured at relatively low concentrations of perchlorate. However, they should be careful not to imply that $n = 2$ allows extrapolation to all mesophilic proteins (e.g., Results, 2nd paragraph, top of p3 “To further validate...” and top of p5).

Author response: We thank the reviewer for the suggestion and have removed this language (p3) and adjusted it (p5) to avoid implying that $n=2$ is all-encompassing.

6. p3 – Sigmoidal traces do not indicate “proper folding,” as stated, but rather ‘cooperative folding.’ It is safe to say that these are still folded molecules.

Author response: We thank the reviewer for noting this. Because the trace is reporting on DFHBI fluorescence, which is negligible in the absence of a correctly folded RNA, it is reporting on correct folding of the aptamer. We have updated the text to increase clarity in this respect.

7. Fig 1k. Main polymerization product appears to be at +8 relative to primer. y-axis of next panel (Fig 1l) refers to “Product >7nt”. Primer itself appears to be 10nt, with 18 additional nucleotides available for polymerization (from Extended Table 1). Are there additional polymerization products beyond +8 (gel image is cut off above that point)? At a minimum, address these points in the text; probably also prudent to show uncropped or less cropped image in SI.

Author response: We thank the reviewer for the comment. There are three additional nucleotides beyond the $n+8$ product shown in the figure, and we based our initial analyses on the $n+8$ product because this is the furthest primer extension product in most gels observed. In our revised manuscript, we have included gels showing the full suite of products through $n+11$ in the main text, and we have included uncropped gel images in the SI.

8. Fig 3 shows readthrough of a short (3bp) hairpin by polymerase ribozyme in the presence of chlorate. Two points: a) Have the authors looked at longer hairpins or at stem-loops with different stabilities (e.g., one or more GC pairs or capped by a stable tetraloop)? What is the upper limit for this helicase-like effect? b) Graphics above panels e and f are misleading. They look like two different templates are used in the two experiments, when the intention is to depict unwinding of the inhibitory 3bp ‘roadblock.’ Please adjust image or state explicitly in the legend what the graphics represent.

Author response: We thank the reviewer for the suggestion. Based on extensive experiments with longer/GC-containing stem-loops, we have observed that primer extension through these more challenging templates does not occur. Accordingly, we have revised the figure and section discussing perchlorate-associated benefits to relieving secondary structure-associated inhibition to focus on hammerhead/rN6 data in the revision. We feel that this use of the mild denaturant properties of perchlorate best showcases this phenomenon, and we appreciate the opportunity to clarify this point.

9. p9, bottom. To provide context for understanding the HRP-catalyzed reaction, state (and show data for) chlorite concentration that denatures HRP. Optional: Evaluation of HRP stability in chlorite could be shown in conjunction with protein denaturation in Fig 1, either as additional panels for Fig 1 or within SI.

Author response: We thank the reviewer for the suggestion and have added these data to Supplementary Figure 8.

10. Fig 4e, these reactions were performed in the presence of chlorate, correct? Give concentration.

Author response: We thank the reviewer for the comment. These were not done in the presence of perchlorate (chlorate was not used throughout the manuscript). We have added data to the supplementary information showing rPS2.M/hemin exhibited catalytic activity up to 5-6M perchlorate (SI figure 8).

11. Turnover calculation is based on data shown in ED Fig 6. Indicate how “50 μ M MCD” is calculated from these data; if different data were used to arrive at that value, then show the data from which that number is calculated.

Author response: We thank the reviewer for their comment. The 50 μ M MCD concentration given was an experimental parameter, and it was not calculated. The number calculated in this figure is 12.5 peroxidation cycles on the basis of the required number of equivalents of chlorite and its concentration. This is described in SI Figure 9 of the revised manuscript.

12. ED Fig 7 does not communicate well. Panels (a) and (b) appear identical, except for (b) being slightly fuzzier, making it difficult to discern a meaningful interpretation. In panel d, point to expected product (near 381-383 units) as in Fig 4i-j and give expected mass.

Author response: We appreciate the reviewer's suggestions to improve clarity in our manuscript. We have added an arrow to highlight the expected products and updated the text to provide the relevant information to better communicate these results. Panels A and B are intended to indicate that hemin alone does not significantly enhance

chlorination, and that the addition of the PS2.m RNA is required. We have moved two of the figure panels from Figure 4 to SI Figure 10 in the revised manuscript to better communicate this information and avoid confusion.

MINOR DETAILS

- When the object of a verb is a clause, the clause needs to be introduced by “that.” For example, “I know THAT this is true.” rather than “I know this is true.” (“This is true” is a clause.) Some examples from this manuscript:
- Second line of abstract: “indicating that oxychlorine species”
- Sixth line of Intro: “suggesting that near-surface”
- Fifth line of Results: “We found that the hammerhead”
- Third paragraph of “An RNA-heme holoenzyme” section, fifth line: “We observed that these reactions”
- Near end of next paragraph: “These results indicate that the rPS2.M/heme holoenzyme”
- Near end of second paragraph of Discussion: “our results show that RNA is innately resistant”
- Two paragraphs later: “The RNAs we have studied show that resistance to”
(Note that penultimate paragraph already uses this more proper structure: “In any such scenario, our results ... suggest that Martian molecular evolution...”)
- Second sentence of last paragraph of Introduction needs a verb. (Starts “Particularly, ...”)

Author response: We thank the reviewer for these suggestions and have made the requested corrections.

p2, Results, first paragraph. Hyphenate “first-order”. Suggestion: Remove comma and reword as “and retention of function in brines...” Suggestion for last sentence: “Thus, while the highest concentrations of perchlorate somewhat suppressed ribozyme function, robust catalysis was still permitted at remarkably high concentrations.”

Author response: We thank the reviewer for these suggestions and have made the requested corrections.

p2, Results, second paragraph. First two sentences: “...we examined was derived from the avocado sunblotch viroid. The host organism that it infects, avocado (*Persea americana*) is mesophilic and is among the most salt-sensitive crops, ...”

Author response: We thank the reviewer for these suggestions and have made the requested corrections.

p3, second full paragraph. Suggestion: “Ligand binding (exemplified by Broccoli

aptamer) and ...” Five lines later: Odd phrasing in “This was slightly more than half the yield,” since the preceding sentence was about $d(\text{yield})/d[\text{anion}]$. “This” is ambiguous.

Author response: We thank the reviewer for these suggestions and have made the requested corrections.

Fig 1 legend. Capitalize “Graphical” in text for panel f.

Author response: We thank the reviewer for these suggestions and have made the requested corrections.

p5, near the bottom. Suggestion: “In contrast, all three RNAs studied recovered activity upon dilution...” (It took me a while to realize that the three bars represent three different RNAs rather than a progressive dilution from high chlorate to low chlorate. Explicitly saying “three” could help with this.)

Author response: We thank the reviewer for these suggestions and have made the requested corrections.

p7, second paragraph. Use plural (“hexamers”) throughout, since there are 4096 of them.

Author response: We thank the reviewer for these suggestions and have made the requested corrections.

p9, second paragraph. Show charges for chlorite and hypochlorite; these are currently shown as neutral species! Cite reference for “Job’s method.”

Author response: We thank the reviewer for these suggestions and have made the requested corrections.

p10, top. Use singular: “To our knowledge, this is the first report of carbon-chlorine formation...”

Author response: We thank the reviewer for these suggestions and have made the requested corrections.

p11, Discussion. Suggestion for first sentence: “...evidence demonstrating that RNA is uniquely well-suited ... brines (no comma) like those thought to occur...” 2nd and 3rd paragraphs: Add comma before WHICH: (“which can grow in solutions...” and “which disrupts the hydrophobic collapse...” End of 2nd paragraph: It would safe to omit “periods.” 4th paragraph, change “like thermophilic proteins” to “similar to that of ...”

Author response: We thank the reviewer for these suggestions and have made the requested corrections.

p18 Methods. Hammerhead Kinetic Assay. Is 75% ethanol the final concentration or (as written) the concentration of the material that was added? Next paragraph: Were strands lyophilized together or separately?

Author response: We thank the reviewer for noting this. We have clarified the methods section to indicate that a final concentration of 75% ethanol was present, and that only strand B was lyophilized and strand A was added during resuspension of strand A with addition of salts.

p19. Ribozyme polymerase assay. Optional: It would be safe to omit “out of solution.” Two paragraphs later, add comma after “nitrocefin” in first line. Next paragraph, specify what samples were used as “positive control and negative control”

Author response: We thank the reviewer for these suggestions and have made the requested corrections.

EXTENDED DATA: Define meanings of error bars and number of independent observations (n) on which they are based.

Author response: We thank the reviewer for this suggestion and have made the requested change.

ED Fig 3. What equation was used to fit these data? Indicate where the inferred midpoint transitions (and T_m values) lie. It is difficult to infer an accurate T_m from only 5 data points! Even if these data were not used to calculate T_m values (which appears to be the case), it looks like the data points are simply connected with a cubic spline so that the curves pass exactly through the points.

Author response: We thank the reviewer for their attention to detail on this figure. The data shown were collected at 1 °C data pitch. The data points shown give representative error bars and differentiate the curves with symbols. The curve shown is a sigmoidal fit of all these data points; we have updated the manuscript to reflect this.

ED Fig 5. In x-axis, denominator needs parentheses

Author response: We thank the reviewer for this suggestion and have made the requested change.

We hope with these changes to the manuscript, along with those we have made in response to suggestions from the other two reviewers, that the reviewer will now find our manuscript suitable for publication in Nature Communications.

Reviewer #2 (Remarks to the Author):

Hoog et al. present a survey of the comparative performance of functional RNAs and proteins in perchlorate brines, as a model for functional biomolecules in similar brines that could form on the surface or sub-surface of Mars in the latter dry stages of that planet's geological evolution. Functional RNAs investigated include the hammerhead ribozyme, Spinach fluorogenic aptamer, and tC19Z RNA polymerase ribozyme. The proteins investigated include nucleases EcoRI, RNaseHII, and TaqI-v2, the former two from mesophilic *E. coli* bacteria, the latter from the thermophilic *T. aq.* bacteria, and a β -lactamase from a halophilic bacterium.

The authors reach the conclusion that RNA is broadly and generally tolerant of concentrated perchlorate, with each RNA tested able to tolerate at least 2 M and as much as 6 M perchlorate. Proteins, in contrast, cannot tolerate perchlorate above 0.5 M, with the exception of a halophile derived β -lactamase. The authors ascribe this difference, primarily, to the polyanionic nature of RNA that would be resistant to salt-induced denaturation, while proteins must evolve negatively charged surfaces or extreme stability, as occurs in halophilic or thermophilic proteins, respectively, to achieve comparable levels of resistance.

The ability of RNA to maintain function in concentrated perchlorate is shown to enable further advantages. Complex mixtures of RNA oligonucleotides are disruptive to RNA function, by binding of oligonucleotides to functional regions of an RNA structure. In a model of such disruption using the hammerhead ribozyme, it is shown that concentrated perchlorate can significantly reduce this inactivation, presumably by destabilizing interstrand RNA pairing. Likewise, modest levels of perchlorate appear to improve polymerase read-through of an RNA template with a central stem-loop structure that would normally block the polymerase, suggesting that mild denaturation of RNA structure in perchlorate brines could improve RNA-catalyzed replication of structured RNA.

Finally, and separately, the horseradish peroxidase protein enzyme and the rPS2.M ribozyme, both of which use heme as a catalytic cofactor, are shown to chlorinate various compounds from chlorite, another known component of Martian soils, at comparable rates. Chlorination appears to be regiospecific, can occur as a multi-turnover reaction, and can produce polychlorinated products. These results are taken as preliminary evidence that either biopolymer could evolve to incorporate environmental oxychlorine into metabolism.

Overall, the authors present compelling, albeit preliminary, evidence that RNA is intrinsically better suited to maintain function in concentrated perchlorate solutions, and, indeed, that these brines may offer specific advantages for RNA catalysis. The authors hypothesize that functional RNA would have been favored by evolution in perchlorate brines over functional proteins, either as an emergent form of life from prebiotic

chemistry or an adaptive response of extant Martian life as the Martian surface dried to its current state. This is a highly valuable approach to origin of life investigations, tying biochemical behavior to prebiotic environments on Earth and elsewhere, in this case one that is decidedly hostile to extant biology, and thus not explored in detail by biochemists focused on the biological behavior of macromolecules. These results will be of general interest to RNA biochemists, prebiotic chemists, and astrobiological fields more broadly, and is highly suitable for publication in Nature Communications.

Author response: We are pleased the reviewer finds our evidence compelling, that our work represents a highly valuable approach to origin of life investigations, and that our results will be of general and broad interest and are highly suitable for publication in Nature Communications. We have revised the manuscript in response to their specific criticisms, which we detail below.

There are three points that I think should be addressed before publication, however. Each can be addressed by minor revisions to the text alone, although additional experimental work may increase the confidence of some conclusions. First, nuclease activity is the only shared enzyme type for both protein and RNA enzymes in this survey. However, it is unclear if these were assayed under comparable kinetic regimes. The hammerhead ribozyme was assayed under more-or-less single turnover kinetics, with the two pieces of the ribozyme (one strand of which is the substrate RNA that is cleaved) present in equimolar amounts. It is unclear if the protein nucleases were assayed under similar conditions, as the enzyme input to these reactions is given as units of activity, not molarity. It is sometimes difficult to determine protein concentration in enzyme stocks supplied commercially, but it should be possible for at least some of these enzymes. The manuscript should clarify if the kinetic regimes for RNA and protein were similar, and if not, the implications for differing effects due to different rate-limiting steps: bond cleavage vs product on/off rates etc. As a broad survey of activity, a perfect match between kinetics isn't required, but the consequences of any differences should be discussed. It would also be helpful if some basic details of the kinetics (enzyme and substrate stoichiometry, pH or temperature where relevant) for each reaction were provided in the results or relevant figure legends, rather than only in the methods.

Author response: We thank the reviewer for this suggestion. In response, we have added reaction conditions to each figure. We also have updated the figure legends to supply additional information on reaction conditions. We have additionally expanded on the experimental details of enzyme assays in each figure and have included a discussion of the consequences of the differences of the assays employed as the reviewer suggests. We agree that this additional clarification of our assays will be helpful to the reader.

Second, a relatively minor point, but the difference in recovery of activity after exposure to high perchlorate concentrations is compared between the hammerhead ribozyme, broccoli aptamer, and tC19Z polymerase ribozyme in the last paragraph of page 5. The

greater recovery for the polymerase may be illusory. From the methods, it appears that the polymerase ribozyme was assayed by yield at a single time point, rather than measuring an observed rate constant, as was performed for the hammerhead ribozyme. If the reaction is near its maximum extent when the timepoint is taken, rather than in the linear phase, differences in yield from different conditions can be lower than difference in the underlying rate of reaction, as faster reactions will be approaching the maximum extent limit. All three RNAs recover significantly after exposure to perchlorate brines, but the text should not imply that the greater measured recovery for the polymerase is notable, unless a more detailed measurement of polymerase rates is performed.

Author response: We thank the reviewer for the suggestion and have adjusted the wording in this section.

Third, the fact that perchlorate brines can “loosen” RNA structure, while enabling RNA enzymes to retain function, is extremely interesting, and as noted, important for the copying of structured RNA by polymerase ribozymes. The results reported compare yield of extension products past a 3 nucleotide stem loop in an RNA template in chloride or perchlorate solutions. Sodium is known to be detrimental to the polymerase (J. Attwater, et al., Chemical fidelity of an RNA polymerase ribozyme. *Chem. Sci.* 4, 2804–2814 (2013)), and polymerases can perform idiosyncratically with different templates under different conditions. Also, the sequence of the template stem is AU rich, which is generally copied less efficiently by the polymerase. Although it seems likely that the better performance in perchlorate is due specifically to disruption of the stem loop, it could alternatively be due to perchlorate ameliorating the effects of sodium, aiding in AU copying, or other challenges. Experiments with other templates, either varying stem loop sequence, or including a negative control where the stem loop is disrupted but the ~7 nt portion copied by the polymerase is the same sequence, would provide a much more convincing demonstration that perchlorate is specifically aiding RNA copying by disrupting secondary structure in the template. In the absence of that data, since this is not a central experiment to the paper, it would be sufficient to change the language to make clear the tentative nature of the conclusions regarding the effects of perchlorate on secondary structure.

Author response: We thank the reviewer for the suggestion. Based on this suggestion and that of reviewer #1, we tested other stem-loop structures. In these experiments, we observed that we cannot obtain primer extension through these more structured templates. Accordingly, we have chosen to refocus this section on the hammerhead activation data, as we feel these data better showcase the potential of the mild denaturant properties of perchlorate to activate latent functions in RNA. Results from an experiment suggested by reviewer #1 further bolster this decision. In this experiment, we performed experiments with the hammerhead ribozyme in urea, and found that it was not compatible with activation in the presence of rN6, presumably because it was too strong a denaturant, and that perchlorate is in a “sweet spot” of mild denaturation in

this system. Given these data, we have opted to focus this section on this aspect of perchlorate-functional RNA interactions.

We hope with these changes to the manuscript, along with those we have made in response to suggestions from the other two reviewers, that the reviewer will now find our manuscript suitable for publication in Nature Communications.

Reviewer #3 (Remarks to the Author):

Emergent functional behaviors of ribozymes in oxychlorine brines indicate Mars could host a unique niche for molecular evolution
Hoog TG et al.

Major comments

The paper presents some very interesting work on the effects of perchlorate on biomolecular function. In particular the description of new capabilities associated with nucleic acids is of great interest. The paper is well-written and very clear. I only have some minor comments.

Author response: We are pleased the reviewer finds our work of great interest, well-written, and clear. We have revised the manuscript in response to their comments, which we detail below.

One point: It's a bit strange that the paper doesn't cite any of the previous papers that present information on the effects of perchlorate brines on biomolecular function relevant to Mars. There are these papers for example:

<https://pubs.acs.org/doi/abs/10.1021/jacs.1c01832>

<https://www.nature.com/articles/s41598-021-95997-2>

<https://www.nature.com/articles/s42003-020-01279-4>

Although the work is different, the field of research that deals with perchlorate brines on Mars and effects on biochemical function is very niche and it would be worth acknowledging these, especially those that show improvements in function in perchlorate (for example pressure reversal of deleterious perchlorate effects).

Author response: We thank the reviewer for the suggestion, and we have included these citations in the introduction to better contextualize our work. We appreciate the suggestion and agree that these enhance the manuscript.

Minor comments

Page 5, line 3. I don't think it is necessary to point out that E. coli is terran – could be removed.

Page 5, line 3. It's a bit of a stretch to say that these enzymes could have been found in a Noachian/Hesperian organism as we don't know if Mars ever had life. Maybe this

could be toned down to simply say that these enzymes could be plausible universal analogues of any organism.

Page 7, line 4 up. The statement that increases in salt concentrations did not result in further elongation product, but there was still an increase in elongated could be clearer.

Page 9, line 7 up. Delete period and extra space after 'chlorination reactions'.

Under 'restriction enzyme kinetic assay. Line 9 down. Remove extraneous parenthesis.

Figure 4 is quite dense. Maybe some of the more methodological parts could go in supplemental?

Author response: We thank the reviewer for the suggestions and have made corrections to the text in each case. Additionally, we have moved two of the panels from Figure 4 to the supplementary information.

We hope with these changes to the manuscript and those in response to the suggestions of the other two reviewers, that the reviewers and editor will now find our manuscript suitable for publication in Nature Communications.

REVIEWERS' COMMENTS

Reviewer #1 (Remarks to the Author):

The original manuscript was fascinating and mostly well done. The authors have responded well to suggestions and applied the appropriate polish. I am satisfied with this product and ready to see it move forward.

A few specifics (None of these are negatives; none needs a reply):

1. ✓

2. New data are nice.

3. ✓

4. ✓

5. ✓

6. ✓ (Well, sort of. The authors are correct that the fluorescence data indicate “proper folding.” However, the shape of the curve (“sigmoidal”) indicates that the unfolding happens cooperatively, not that it is functional. However, taken together and in context, I am ok with the revised wording.)

7. ✓

8. ✓

9. ✓

10. ✓

11. ✓

12. Ah. I get it. Thanks. Revised figure is nice.

OTHER DETAILS. ✓

Reviewer #2 (Remarks to the Author):

The authors have satisfactorily addressed the points brought up in my previous review. The authors' updated details on reaction conditions and the expanded discussion of these conditions in the discussion is appreciated, and I broadly agree that the conclusions still hold despite the different conditions used for each reaction. Future investigation of the detailed

kinetics of both RNA and protein enzymes, may be quite fruitful in elucidating general principles for the effects of perchlorate on catalytic activity and folding with both polymers.

The authors' decision to remove discussion of polymerase ribozyme strand invasion behavior, based on experiments with more stable stem-loop structures, and focus on the more robustly supported hammerhead activation results is reasonable. Additional experiments showing that hammerhead activation in the presence of mixtures of oligonucleotides cannot be achieved with a neutral denaturant like urea supports the conclusion that perchlorate is providing a unique benefit to RNA folding and function. Despite the inability of the polymerase ribozyme to extend through more stable stem-loops, the initial strand invasion experiments were promising. It may be worth revisiting this phenomenon in future work using polymerase ribozymes with more robust strand invasion activity or processivity (Cojocaru & Unrau, *Science* 2021; Attwater et al, *eLife* 2018; Horning & Joyce, *PNAS* 2016).

As stated in my earlier review, I believe the manuscript will be of broad interest to researchers in the origin of life and astrobiology fields and is suitable with the revisions provided by the authors for publication in *Nature Communications*.